# Transcriptome Profiling Unveils Key Genes Regulating the Growth and Development of Yangzhou Goose Knob

**DOI:** 10.3390/ijms25084166

**Published:** 2024-04-10

**Authors:** Xinlei Xu, Suyu Fan, Wangyang Ji, Shangzong Qi, Linyu Liu, Zhi Cao, Qiang Bao, Yang Zhang, Qi Xu, Guohong Chen

**Affiliations:** 1Key Laboratory for Evaluation and Utilization of Poultry Genetic Resources of Ministry of Agriculture and Rural Affairs, Yangzhou University, Yangzhou 225009, China; xuxinlei0914@163.com (X.X.); fansuyu20030729@163.com (S.F.); wangyangji1995@163.com (W.J.); qsz8200@163.com (S.Q.); liulinyu9812@163.com (L.L.); cz102911@163.com (Z.C.); dx120200144@stu.yzu.edu.cn (Q.B.); xuqi@yzu.edu.cn (Q.X.); ghchen2019@yzu.edu.cn (G.C.); 2Joint International Research Laboratory of Agriculture and Agri-Product Safety, The Ministry of Education of China, Yangzhou University, Yangzhou 225009, China

**Keywords:** DEGs, knob, RNA-seq, WGCNA, Yangzhou goose

## Abstract

Goose is one of the most economically valuable poultry species and has a distinct appearance due to its possession of a knob. A knob is a hallmark of sexual maturity in goose (*Anser cygnoides*) and plays crucial roles in artificial selection, health status, social signaling, and body temperature regulation. However, the genetic mechanisms influencing the growth and development of goose knobs remain completely unclear. In this study, histomorphological and transcriptomic analyses of goose knobs in D70, D120, and D300 Yangzhou geese revealed differential changes in tissue morphology during the growth and development of goose knobs and the key core genes that regulate goose knob traits. Observation of tissue sections revealed that as age increased, the thickness of the knob epidermis, cuticle, and spinous cells gradually decreased. Additionally, fat cells in the dermis and subcutaneous connective tissue transitioned from loose to dense. Transcriptome sequencing results, analyzed through differential expression, Weighted Gene Co-expression Network Analysis (WGCNA), and pattern expression analysis methods, showed D70-vs.-D120 (up-regulated: 192; down-regulated: 423), D70-vs.-D300 (up-regulated: 1394; down-regulated: 1893), and D120-vs.-D300 (up-regulated: 1017; down-regulated: 1324). A total of 6243 differentially expressed genes (DEGs) were identified, indicating varied expression levels across the three groups in the knob tissues of D70, D120, and D300 Yangzhou geese. These DEGs are significantly enriched in biological processes (BP) such as skin morphogenesis, the regulation of keratinocyte proliferation, and epidermal cell differentiation. Furthermore, they demonstrate enrichment in pathways related to goose knob development, including ECM–receptor interaction, NF-kappa B, and PPAR signaling. Through pattern expression analysis, three gene expression clusters related to goose knob traits were identified. The joint analysis of candidate genes associated with goose knob development and WGCNA led to the identification of key core genes influencing goose knob development. These core genes comprise *WNT4*, *WNT10A*, *TCF7L2*, *GATA3*, *ADRA2A*, *CASP3*, *SFN*, *KDF1*, *ERRFI1*, *SPRY1*, and *EVPL*. In summary, this study provides a reference for understanding the molecular mechanisms of goose knob growth and development and provides effective ideas and methods for the genetic improvement of goose knob traits.

## 1. Introduction

Goose is a common poultry breed that plays an important role in the supply of animal meat [1]. The World Health Organization considers goose meat an important “healthy food”. China is the world’s largest goose-raising country, with production reaching 4.29 million tons in 2021 [2]. Apart from the Ili goose, the origin of Chinese geese can be traced back to the Swan goose (*Anser cygnoides*), with 30 different local breeds [3]. Compared with European geese, Chinese geese have more slender necks. In adulthood, a fleshy protuberance, otherwise known as the knob, grows at the top of the goose beak [4]. The knob is not only a characteristic sign of sexual maturity in Chinese geese, it additionally provides crucial information for artificial breeding and disease observation. By observing the knob, factors such as the sex, age, and size of the goose can be identified, making it a key factor in goose sales [5,6,7]. Therefore, knobs are considered a secondary sexual characteristic of geese. However, the genetic mechanisms underlying this knob phenotype have not yet been elucidated.

Extensive studies on the genetic mechanisms of head tissues have been conducted in other poultry species. In chickens, the comb is a fleshy protuberance found on the top of the head [8], and significant differences in egg-laying performance and growth rate were observed between Pea Comb (PC) and Single Comb (UC) chickens [9,10]. The genes *BMP2* and *CHADL* are related to chicken comb size, whereas *STK32A*, *PIK3R1*, and *EDN1* regulate the development of the chicken comb [11]. Additionally, genomic duplication is associated with ectopic eomesodermin expression in embryonic chicken combs and two duplex comb phenotypes [12]. In crested ducks, the *Hoxc8* gene has been found to be closely related to feather growth and skull development [13]. Intermating between crested ducks may result in offspring with upper beak and skull deformities, brain protrusion, and skull-pharyngeal parasites manifesting in the form of underdeveloped legs [14]. Furthermore, the head feathers of male wild ducks appeared green, whereas females exhibited a subdued head feather color. Genes such as *TYR*, *SLC38A11*, *REELD1*, and *SYNPR* may influence the distinct color differences between the head and back feathers of males and the head feathers of females, with *TYR* and *TYRP1* being directly related to melanin biosynthesis [15]. Knobs represent a unique appearance characteristic of geese, but their tissue morphology and genetic mechanisms have not yet been revealed. A comprehensive analysis of Lander, Lionhead, and Sichuan white geese using histomorphology, transcriptome, and whole-genome resequencing has, for the first time, unveiled the possibility that *DI02* plays a key role in the phenotype of goose knobs. [16]. Through phenotypic observation, it was found that Yangzhou geese with larger knobs at the same age tended to be heavier and more favored by consumers. Therefore, a joint analysis of histomorphology and transcriptomics was conducted to identify *BMP5*, *DCN*, *TSHR*, and *ADCY3* as key genes influencing the development of the Yangzhou goose knob [17]. These findings indicate that studying head phenotypic features is crucial for understanding growth rate, skull development, and disease observation. Although studies have identified important genes that may affect the phenotype of goose knobs, there are currently only the above two relevant reports on the genetic mechanisms affecting goose knobs. Therefore, the genetic mechanisms affecting the growth and development of goose knobs cannot be fully revealed. However, as goose is one of the most economically valuable poultry, it is of great significance to study the genetic basis of goose knobs.

An increasing number of studies have revealed the genetic basis of multiple species’ phenotypic traits through comprehensive analyses of phenotypes, tissue morphology, and transcriptome sequencing data [16,17,18]. Therefore, this study aimed to uncover differential changes in tissue morphology during the growth and development of goose knobs. We also sought to identify key core genes that regulate goose knob traits, laying the theoretical foundation for further understanding of the genetic basis of Yangzhou goose knobs.

## 2. Results

### 2.1. The Histomorphological Analysis of Knob Skin in Geese

To reveal the histological structure of the knob skin, we cut the knob skins of D70, D120, and D300 Yangzhou geese along the horizontal axis into sections and stained them. Upon scrutinizing the tissue sections (Figure 1, Appendix A), it was evident that the skin of the goose knob was divided into the epidermis (stratum corneum, stratum granulosum, and stratum spinosum), dermis (consisting of papillary and reticular layers), and subcutaneous tissue from the outside to the inside. With age, discernible variations manifest in the structure of the epidermis, dermis, and subcutaneous connective tissue in the anserine knob skin morphology. From D70 to D300, there is a gradual reduction in the thickness of the epidermis, stratum corneum, and spinous cells in the goose knob. Simultaneously, adipocytes in the dermis and subcutaneous connective tissue experience a gradual proliferation, reaching a plumper and denser state by D300. During this period, the development of the goose’s knob steadily advances, leading to a densely intertwined fiber bundle network characterized by heightened toughness and elasticity.

### 2.2. Goose Knob Development-Related Transcriptome Sequencing

To elucidate the regulatory mechanisms governing the growth and development of the Yangzhou goose knob, we isolated total RNA from nine samples collected at D70, D120, and D300. A transcriptome library was generated and sequenced. Quality control assessments revealed that the Clean Data for each Yangzhou goose knob sample exceeded 6.81 Gb, with a Q30 ratio exceeding 93.56% and a GC content ranging between 49.32% and 50.78% (Appendix A). The sequencing quality was reliable, affirming its suitability for subsequent data processing. The PCA results indicated significant differences among the D70, D120, and D300 Yangzhou goose knobs (Figure 2A). Additionally, we performed statistical analyses of the distribution of gene expression across the nine samples, as represented by box and density plots (Appendix A). Employing the DESeq2 package with |log2FC| ≥ 1 and P adjust < 0.05 as screening criteria, we identified DEGs: D70-vs.-D120 (up-regulated: 192; down-regulated: 423), D70-vs.-D300 (up-regulated: 1394; down-regulated: 1893), and D120-vs.-D300 (up-regulated: 1017; down-regulated: 1324). This resulted in 6243 DEGs (Figure 2B and Appendix A). Notably, 82 DEGs (up-regulated: 30; down-regulated: 52) were common among the three groups (Figure 2C). Cluster analysis on the DEGs within the three groups (Figure 2D) revealed that samples within the same group clustered together, further validating the accuracy of the sequencing results.

To explore the biological functions affecting the growth and development of the Yangzhou goose knob, Gene Ontology (GO) functional analysis and Kyoto Encyclopedia of Genes and Genomes (KEGG) pathway analysis were conducted on the 6243 DEGs. In the Biological Process (BP) category, DEGs were primarily associated with immune response, cell chemotaxis, and skin morphogenesis. In the Cellular Component (CC) category, DEGs were mainly related to the extracellular space, integral components of the membrane, and intrinsic components of the membrane. In the Molecular Function (MF) category, the DEGs were predominantly associated with immune and cytokine receptor activity (Appendix A). DEGs were significantly enriched in signaling pathways, such as TNF, MAPK, cAMP, PI3K-Akt, ECM–receptor interaction, NF-kappa B, and PPAR (Appendix A). It is worth noting that previous studies have also observed significant enrichment in pathways like PI3K-Akt, PPAR, ECM–receptor interaction, cytokine–cytokine receptor interaction, and cAMP, indicating the crucial roles of these pathways in the growth and development of goose knob [16,17]. To further elucidate the functions of the DEGs, GO (mainly focusing on BP terms) and KEGG pathway enrichment were performed on the adjacent control groups (Figure 3A,B, Appendix A). For D70-vs.-D120, significant enrichment was observed in terms of signaling receptor binding, the extracellular matrix, the protein folding chaperone, the regulation of epidermal development, skin morphogenesis, the BMP signaling pathway, the establishment of a skin barrier, and other terms related to skin development. GO terms for D70-vs.-D300 include the immune system process, the intrinsic component of the membrane, and the integral component of the membrane. D120-vs.-D300 is mainly enriched in B cell activation, lymphocyte activation, leukocyte activation, and terms related to skin and keratinocyte differentiation, such as skin morphogenesis and mesenchymal-epithelial cell signaling. In summary, DEGs on D70-vs.-D120 enrich in multiple terms related to epidermal development, suggesting a direct association with skin growth and development. DEGs in D70-vs.-D300 are related to the immune response and membrane composition. D120-vs.-D300 mainly enriches cell activation. The MAPK, cAMP, ECM–receptor interaction, PI3K-Akt, and PPAR signaling pathways were predominantly enriched in all three control groups.

To explore the potential key genes associated with goose knob development, we used Cytoscape software to visualize the DEGs related to goose knob development in the GO and KEGG pathways. The enriched BP terms included skin morphogenesis, the regulation of keratinocyte proliferation, epidermal cell differentiation, the positive regulation of keratinocyte proliferation, the canonical Wnt signaling pathway, keratinocyte development, and the regulation of epidermal growth factor-activated receptor activity. The related genes included 27 (*SLC38A26*, *SLC38A25*, *ANXA8L156*, *WNT11*, *HPSE272*, *WNT4*, *WNT7A*, *WNT10A*, *TCF7L2*, *WNT2B*, *GATA3*, *FZD1*, *WNT3A*, *BCL9*, *ADRA2A*, *VPS25*, *CASP3*, *ANXA1*, *SLC35G138*, *FGF7*, *NOTCH2*, *SFN*, *PPRC191*, *PRKD1*, *CEP5570*, *KDF1*, *ERRFI1*) genes (Figure 4A). Similarly, in the pathway analysis, 15 genes (*WNT11*, *HPSE272*, *WNT4*, *WNT7A*, *WNT10A*, *TCF7L2*, *WNT2B*, *FZD1*, *WNT3A*, *GATA3*, *NOTCH2*, *ADRA2A*, *CASP3*, *FGF7*, *PRKD1*) were enriched in pathways such as the Wnt, PI3K-Akt, neuroactive ligand–receptor interaction, NF-kappa B, Th1 and Th2 cell differentiation, and Rap1 signaling pathways (Figure 4B). These signaling pathways have been confirmed to play a role in regulating skin growth and development across various species, emphasizing the universality of genes involved in these pathways in skin development. Therefore, it can be speculated that they also play key roles in the development of skin tissue in goose knobs. Further analysis of KEGG enrichment results revealed that focal adhesion, JAK-STAT, calcium, T cells, B cells, and other signaling pathways are involved in the growth and development of goose knobs. Similar to the KEGG analysis mentioned above, we conducted a KEGG analysis on pathways potentially associated with goose knob growth and development at different stages, constructing a network diagram based on the genes involved in these pathways (Figure 4C). The results were similar to the aforementioned findings, and numerous genes interacting with the aforementioned signaling pathways were identified as being potentially involved in the growth and development of goose knobs (Appendix A).

### 2.3. Time-Series Expression Analysis

To better understand the changing patterns of gene expression, STEM software was used to perform pattern analysis of the DEG sets in D70, D120, and D300, resulting in eight clusters (Figure 5 and Appendix A). GO analysis was conducted on these eight clusters to identify numerous BP terms related to the tissue types of skin gene expression.

In gene cluster 1, we found that genes are enriched in the regulation of epidermal growth factor-activated receptor activity (*ADRA2A*, *ERRFI1*), the regulation of the epidermal growth factor receptor signaling pathway (*ADRA2A*, *ERRFI1*), the positive regulation of epidermal growth factor-activated receptor activity (*ADRA2A*), the positive regulation of the BMP signaling pathway (*MSX1*, *CYR61*), the epithelial cell morphogenesis (*TNMD*, *STC1*), the positive regulation of fibroblast growth factor production (*WNT2B*), and the regulation of fibroblast growth factor receptor signaling pathways (*SPRY1*, *WNT2B*). Gene expression in this cluster was the highest at D70. We also observed that genes in gene cluster 2 are associated with the regulation of epidermal cell differentiation (*LOC106047235*, *CD109*), the regulation of epidermis development (*LOC106047235*, *CD109*), epithelial cell maturation (*TFCP2L1*), the regulation of epithelial cell differentiation (*LOC106047235*, *CD109*), epithelial cell proliferation (*ACVR2A*, *HGF*), the positive regulation of epithelial cell migration (*ANXA1*, *HSPB1*), skin morphogenesis (*LOC106047235*), the negative regulation of epidermal growth factor-activated receptor activity (*LOC106047235*), and mesenchymal-epithelial cell signaling (*HGF*). Gene expression in this cluster reached its lowest level on D120.

Surprisingly, genes in the third cluster played crucial roles in the regulation of epidermal cell division (*SFN*, *ANXA8L156*), the establishment of the skin barrier (*CEP5552*, *SFN*, *ANXA8L156*), keratinocyte development (*SFN*, *KDF1*), keratinocyte differentiation (*CASP3*, *ANXA8L156*), and epidermal development (*EVPL*, *ANXA8L156*).

### 2.4. Weighted Gene Co-Expression Network Analysis

To explore the intrinsic regulatory mechanisms of mRNA on the characteristics of goose knobs, WGCNA was employed to identify the hub genes influencing the growth and development of goose knobs. After standardizing the gene expression data, an expression matrix consisting of 6,521 genes was constructed (Figure 6A). The heatmap of the expression patterns showed that the yellow (cor = −0.874, *p* < 0.002), blue (cor = 0.875, *p* < 0.002), and brown modules (cor = 0.935, *p* < 0.0002) were strongly correlated with the phenotype groups, including group, length, width, and height. The turquoise (cor = −0.738, *p* < 0.02) and green modules (cor = −0.725, *p* < 0.02) contained genes that were significantly correlated with the phenotype group. Therefore, these five modules were selected for further investigation. GO and KEGG analyses were performed on the genes within these five modules (Pearson correlation ≥ 0.90), revealing significant involvement in biological processes such as the regulation of epithelial cell proliferation, epidermis development, and the regulation of protein tyrosine kinase activity. In the cellular component category, associations were found with integral components of the plasma membrane, replication fork protection complex, desmosomes, and spanning components of the plasma membrane. In the molecular function category, associations were identified between calcium ion binding, armadillo repeat domain binding, enzyme regulatory activity, and G protein-coupled serotonin receptor activity (Figure 6A, Appendix A). Significantly enriched pathways include NF-kappa B, B cell receptor, TGF-β, Rap1, and other important pathways (Figure 6A; Appendix A). Based on the candidate genes associated with knob development (Appendix A), we found that core genes (*WNT4*, *WNT10A*, *TCF7L2*, *GATA3*, *ADRA2A*, *CASP3*, *SFN*, *KDF1*, *ERRFI1*, *SPRY1* and *EVPL*) in the yellow, green, blue, brown, and turquoise modules are involved in the control of knob growth and development processes (Figure 6B).

### 2.5. Pathway Network Regulation of Growth and Development of Goose Knob

By systematically examining the enrichment status of DEGs’ pathways (Appendix A), we identified a total of 17 potential signaling pathways involved in goose sarcoma development, with 167 genes incorporated into the pathway network diagram (Figure 7). Among these, the correlation coefficients of these 167 genes in WGCNA analysis were all greater than or equal to 0.9. From the diagram, it can be observed that all 17 signaling pathways were expressed in D70-vs.-D120, D70-vs.-D300, and D120-vs.-D300. However, only two genes (*AREG* and *MYC*) were expressed in all three time periods (D70, D120, and D300). Interestingly, these two genes are involved in connecting the MAPK- and PI3K-AKT signaling pathways. DEGs uniquely expressed in D70 versus D300 mainly converged in the Ras, MAPK, and Phospholipase D signaling pathways, whereas DEGs uniquely expressed in D120 versus D300 were primarily enriched in the CAMs, PPAR, calcium, neuroactive ligand–receptor interaction, cytokine–cytokine receptor interaction, Th1 and Th2, and NF-kappa B signaling pathways. Genes uniquely expressed at D70 versus D120 were mainly enriched in neuroactive ligand–receptor interactions and neuroactive ligand–receptor interactions that connect the calcium signaling pathway. In conclusion, we found that neuroactive ligand–receptor interactions and calcium signaling pathways play major roles in promoting the occurrence of goose squamous cell carcinoma. NF-kappa B, CAMs, and Th1 and Th2 play crucial roles in processes such as the proliferation and differentiation of goose squamous cell carcinoma epidermal cells. Additionally, the B cell receptor, PI3K-Akt, Focal adhesion, and JAK-STAT signaling pathways are involved in various processes of goose squamous cell carcinoma formation, epidermal cell proliferation, and differentiation.

### 2.6. Quantitative Verification

To verify the accuracy of the KEGG pathway and experimental data, we selected four genes in the pathway related to goose knobs (*BMP5*, *IGF1*, *ITGA11*, *NPPC*) and randomly selected four DEGs (*TEX36*, *ALX1*, *IGFBP4*, *MATN2*) and examined their relative expression using RT-qPCR. The trend of the RT-qPCR results for the eight genes was consistent with that of the transcriptome analysis, confirming the reliability of the transcriptome results (Figure 8).

## 3. Discussion

Relatively less research has been conducted on the histology and genetic mechanisms of goose meat knobs than on head crest characteristics. Long-term observations of the experimental population showed that the meat crests of Yangzhou geese did not form during the embryonic period. Changes at the crest base began in chicks at 70 days of age, developing into a hard, conical bony prominence at 120 days of age, with no significant changes until 300 days of age. Therefore, this study comprehensively analyzed and compared the histological and transcriptomic characteristics of Yangzhou goose meat knob skin tissues on D70, D120, and D300.

Histological analysis revealed that the thicknesses of the epidermis, stratum corneum, and spinous layer cells of the goose meat knob gradually decreased from D70 to D300. Fat cells in the dermis and subcutaneous connective tissue gradually proliferated from D70 to D120 and became more abundant and compact by D300. These results indicated significant differences in the structure of the epidermis, dermis, and subcutaneous connective tissue of goose meat knobs from D70 to D300. Therefore, understanding the genetic mechanisms of goose meat knob development is crucial for the genetic improvement of the breed.

By identifying the expression patterns of the DEGs, three gene clusters (Cluster 1, Cluster 2, and Cluster 3) that may affect the growth and development of goose knobs were obtained. Cluster 1 genes (*ADRA2A*, *ERRFI1*, *MSX1*, *CYR61*, *TNMD*, *STC1*, *SPRY1*, *WNT2B*) and Cluster 2 genes (*CD109*, *TFCP2L1*, *ACVR2A*, *HGF*, *ANXA1*, *HSPB1*) showed similar expression patterns between D70 and D120. However, the expression of Cluster 1 genes after D120 exhibited a significantly smaller increasing trend compared with Cluster 2. This suggests that these DEGs play an inhibitory role in knob growth between D70 and D120. In contrast, the DEGs in Cluster 2 initially played a role in inhibiting knob development and later had a promoting effect on knob development. However, Cluster 3 genes (*SFN*, *ANXA8L156*, *CEP5552*, *SFN*, *KDF1*, *CASP3*, *EVPL*) exhibited expression patterns opposite to those of Cluster 1 and Cluster 2 genes. Among the genes in these three clusters, *ACVR2A*, *HGF*, *ADRA2A*, *HSPB1* were enriched in four pathways of interest: the MAPK signaling pathway, cytokine–cytokine receptor interaction, neuroactive ligand-receptor interaction, and focal adhesion. This further confirms the crucial role of signaling pathways such as cytokine–cytokine receptor interactions and focal adhesion in the growth and development of goose knobs.

At the transcriptome level, 615, 3287, and 2341 DEGs were identified in the comparisons of D70-vs.-D120, D120-vs.-D300, and D70-vs.-D300, respectively. KEGG enrichment analysis of DEGs revealed significant enrichment of signaling pathways such as MAPK, cAMP, PI3K-Akt, ECM–receptor interaction, and PPAR in the D70-vs.-D120 comparison group. In the D120-vs.-D300 and D70-vs.-D300 comparison groups, some genes were enriched in common pathways, including NF-kappa B, CAMs, Th1 and Th2 cell differentiation, and cytokine–cytokine receptor interaction, which were significantly enriched in all three comparison groups.

The PPAR signaling pathway is associated with skin growth and development and plays a key role in determining the phenotype of knobs [16,19]. ECM–receptor interactions are involved in cell morphogenesis, maintenance, and wound repair [20,21,22]. Research has also suggested that the ECM–receptor interaction signaling pathway provides structural support for fat cells and regulates fat production [23], indicating that skin thickening in knobbed goose is regulated by the ECM–receptor interaction pathway. The PI3K-Akt signaling pathway plays a crucial role in regulating the survival and proliferation of keratinocytes [24]. Through large-scale photoacoustic microscopy (LSOM), the overexpression of VEGF-A in transgenic mice during keratinocyte formation of the wound healing process was validated. The results indicated that VEGF was associated with enhanced dermal vascularization in skin wound healing and quantified parameters, such as hemoglobin content, filling fraction, vessel diameter, and tortuosity [25]. The proteomic and transcriptomic exploration of the development of pangolin skin appendages revealed that the MAPK signaling pathway is involved in keratinocyte differentiation, epidermal cell differentiation, and multicellular organism development [26]. By combining previous studies, 17 signaling pathways related to the development of goose knobs, including PPAR, MAPK, VEGF, calcium, cytokine–cytokine receptor interaction, NF-kappa B, and PI3K-Akt, were identified and visualized in the pathway network diagrams. KEGG pathways, as important references for proving the regulation of DEGs, were organized and relevant KEGG pathways were visualized (Figure 9). Additionally, candidate genes such as WNT family members (*WNT11*, *WNT4*, *WNT2B*, *WNT7A*, *FZD1*, *WNT10A*, *WNT3A*) [27], *TCF7L2*, *HPSE272*, *GATA3* [28], *NOTCH2* [29], *ADRA2A*, and *FGF7* [30], related to skin morphogenesis, keratinocyte proliferation, and epidermal cell differentiation, were identified. Finally, through the joint analysis of candidate genes and WGCNA, the key core genes affecting the development of goose knob were identified: *WNT4*, *WNT10A*, *TCF7L2*, *GATA3*, *ADRA2A*, *CASP3*, *SFN*, *KDF1*, *ERRFI1*, *SPRY1*, and *EVPL*.

## 4. Materials and Methods

### 4.1. Ethical Statement

All animal experiments were approved by the Institutional Animal Care and Use Committee of the Yangzhou University (Approval Number: 132-2022). All procedures were performed in accordance with the Regulations on the Management of Laboratory Animal Affairs (Yangzhou University, 2012) and the Standards for the Management of Experimental Practices (Jiangsu, China, 2008).

### 4.2. Animal Sample Collection

The Yangzhou geese used in this study were obtained from Yangzhou Tiange Goose Industry Development Co., Ltd. (Yangzhou, China). After long-term observation at the experimental farm, it was found that the spherical crest of Yangzhou geese did not form during the embryonic period, and changes in the spherical crest began in goslings when they reached 70 days of age. By 120 days, a hard, conical bony prominence had already formed, remaining relatively unchanged until 300 days of age (Appendix A). Therefore, 500 healthy male Yangzhou geese were selected from one-day-old goslings for the experiment and raised using a standard feeding protocol. At 70 (D70), 120 (D120), and 300 (D300) days of age, three male geese were selected from each of the three experimental goose groups, and arterial blood was drawn from the neck to ensure euthanasia. The apical integumentary outgrowth (including skin and subcutaneous connective tissue) tissues of the knobs were collected and promptly preserved in liquid nitrogen to safeguard their integrity, and the collection of their phenotypic data occurred (as the knobs had not distinctly formed by D70, the phenotypic data were exclusively documented at D120 and D300). Finally, they were stored in a freezer (−80 °C) for subsequent RNA isolation. Additionally, integumentary outgrowth tissues were collected for paraffin sectioning.

### 4.3. Histological Testing

The fixed skin samples were immersed in 4% paraformaldehyde and allowed to rest for 24 h at 4 °C. Subsequently, the samples were placed in an embedding box, rinsed with running water for 30 min to eliminate the fixative in the tissue, and subjected to a graded ethanol series for moderate dehydration. Paraffin embedding was conducted using a JB-P5 tissue embedding machine (Wuhan Junjie Electronics Co., Ltd., Wuhan, China) at 70 °C. The prepared paraffin blocks were cut along the horizontal axis (RM2016, Germany) into approximately 3 μm thick sections and subjected to staining with hematoxylin and eosin (HE) following standard protocols. The knob skin was examined under an upright light microscope (Nikon, Tokyo, Japan), and image acquisition and analysis were performed using a DS-U3 imaging system (Nikon).

### 4.4. RNA Extraction, Library Preparation, and Sequencing

Total RNA was isolated from skin samples using the TRIzol reagent (Invitrogen, Waltham, MA, USA) following the manufacturer’s instructions. The total RNA purity, concentration, and integrity of each sample were verified using a Nanodrop 2000 instrument (Thermo Scientific, Wilmington, DE, USA) and an Agilent 2100 Bioanalyzer (Agilent Technologies, Santa Clara, CA, USA) (RIN ≥ 8.4, total RNA amount ≥ 1 ug, concentration ≥ 35 ng/μL, 1.8 < OD260/280 < 2.0). RNA sequencing libraries were prepared using the TruSeq™ RNA sample preparation kit from Illumina (San Diego, CA, USA) following the manufacturer’s instructions. In summary, the procedure begins with the use of magnetic beads containing Oligo (dT) and polyA to perform A-T base pairing. Following mRNA isolation (Kb) from total RNA, fragmentation buffer is added to randomly fragment the mRNA, which is then screened through magnetic beads to isolate small fragments of approximately 300 bp. Next, the SuperScript Double-Stranded cDNA Synthesis Kit and six-base random primers (random hexamers) are employed to reverse-transcribe single-strand cDNA from mRNA, followed by second-strand synthesis to form a stable double-stranded structure. Subsequently, End Repair Mix is added to the synthesized cDNA for end repair, followed by addition of base A and sequencing adapters. PCR amplification is then performed after size selection of cDNA target fragments of 300 base pairs (bp) using a 2% low-range ultra-agarose gel. Finally, the amplified fragments were sequenced using an Illumina NovaSeq 6000 sequencer (paired-end 150 bp read length).

### 4.5. Data Quality Control, Comparison, and Assembly

FastQC (Version 0.11.9) software was used to perform quality filtering of the raw data to remove adapter reads, reads with a 10% higher N ratio, duplicate reads, and low-quality reads to obtain clean reads [31]. HISAT2 (Version 2.1.0) software was used to align the sequences with the goose reference genome (*A. cygnoides*, GCF_000971095.1) sequence [32]. The mapped reads for each sample were assembled using StringTie (Version 2.1.2) with a reference-based approach [33].

### 4.6. Bioinformatics Analysis

RSEM (Version 1.3.11) software was used to obtain read counts for each sample gene [34]. TPM con-version was performed to obtain standardized gene expression levels and perform prin-cipal component analysis (PCA) through the gmodels package. The DESeq2 package was utilized to compute DEGs based on the read count expression of each sample [35]. DEGs with an absolute fold change |log2FC| ≥ 1 and an adjusted *p* value (P adjust) of <0.05 were considered to be statistically significant. Excel (2019) was employed to depict the quantitative relationship within each group. The ggVennDiagram package was used to illustrate the intersection genes of DEGs between each group, while the pheatmap package facilitated visualization of the clustering among samples. The GO and KEGG databases were utilized for functional annotation analysis of all sequenced genes, compared against the background of the entire transcriptome. Enrichment analysis of DEGs was con-ducted using the Goatools (Version 0.6.5) and KOBAS (Version 2.1.1) software programs [36,37]. Key pathways were identified through the expression profiles of DEGs within the enriched pathways. Visualization of gene network relationships was achieved using Cytoscape (v3.9.1) software. Furthermore, time-series expression analysis was performed using the STEM (Version 1.3.11) software to observe gene expression patterns across multiple time points. Subsequently, functional class enrichment analysis was conducted on genes exhibiting specific expression patterns to further identify key genes associated with traits [38,39]. The WGCNA package was used to construct a co-expression network for gene expression and the dynamic tree cutting method (setting the MEDiss Thres cutting line to 0.25 to merge similar modules) was employed to divide the modules to determine the final number of modules. Key modules were screened out through |cor| > 0.7 and *p* < 0.02 and correlation analysis was conducted between genes with cor > 0.9 in the key modules and candidate genes related to knob development to determine the key core genes. Cytoscape software was used to visualize all the key pathway networks finally identified related to the development of goose sarcoma. Finally, the regulatory relationships of genes in pathways related to goose sarcoma development were summarized.

### 4.7. Real-Time Quantitative PCR (RT-qPCR)

Total RNA obtained from goose knob skin tissue samples on days 70 (D70), 120 (D120), and 300 (D300) was used for RT-qPCR analysis. DEGs within the relevant pathways identified by RNA-Seq were selected for verification. The cDNA synthesis was carried out using the PrimeScript™ RT kit (Takara, Beijing, China) with the extracted total RNA. RT-qPCR was conducted on an ABI 7500 real-time PCR system (Applied Biosystems, Foster City, CA, USA) using a SYBR PrimerScriptTM RT-PCR kit (Takara). The 20 μL reaction system followed that of the manufacturer’s instructions (three technical replicates per sample), with a program comprising pre-denaturation at 95 °C for 30 s, denaturation at 95 °C for 10 s, annealing at the corresponding temperature for 30 s, and extension at 65 °C for 5 s, repeated for 40 cycles. *GAPDH* served as the internal reference gene, and the 2^−ΔΔCt^ method was employed to calculate the relative expression of genes [40]. We utilized the online primer design software Oligo.7 to design specific primers for eight DEGs (Appendix A).

## 5. Conclusions

We conducted mRNA sequencing at the developmental stages of D70, D120, and D300. Through differential expression analysis, WGCNA, and pattern expression analysis, we identified 2603 up-regulated and 3640 down-regulated DEGs. Additionally, we identified 11 key core genes significantly associated with goose knob development: *WNT4*, *WNT10A*, *TCF7L2*, *GATA3*, *ADRA2A*, *CASP3*, *SFN*, *KDF1*, *ERRFI1*, *SPRY1*, and *EVPL*. By performing functional enrichment and network interaction analysis on DEGs, we unraveled 17 signaling pathways, including Jak-STAT, PI3K-Akt, MAPK, and PPAR, implicated in goose knob development. The roles of these pathways in promoting epidermal morphogenesis and regulating epidermal cells in goose knobs were summarized, emphasizing their significance in proliferation and differentiation processes. In conclusion, this study serves as a reference for understanding the molecular mechanisms underlying goose knob growth and development. It also provides valuable insights and methods for genetically enhancing goose knob traits.

## Figures and Tables

**Figure 1 ijms-25-04166-f001:**
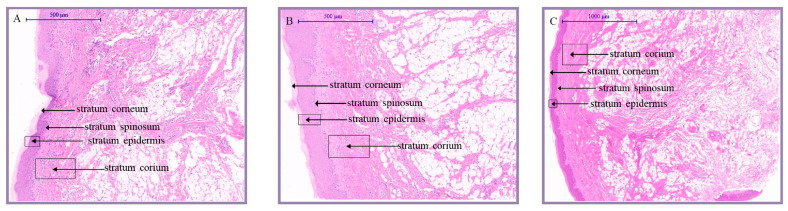
Comparison of histology of Yangzhou goose at different ages. Note: 70-day-old, 120-day-old, and 300-day-old Yangzhou goose knob skin tissues were collected and fixed and stained with HE. (**A**): 70-day-old, (**B**): 120-day-old, and (**C**): 300-day-old knob skin tissue. (**A**,**B**): 7.5×; (**C**): 5×. Between 70 and 120 days of age, the skin exhibited a notably thin structure, while by 300 days of age, it had developed a certain thickness. To accommodate this diversity in skin tissue thickness, we chose to employ two magnifications. The objective is to present the characteristics of the skin at different ages in a more intuitive manner, ensuring that observers can precisely comprehend and compare the structural variations and changes occurring in the skin during distinct periods. This strategic approach aims to offer more comprehensive and lucid image information, facilitating the observation and analysis of subtle changes throughout the process of skin development.

**Figure 2 ijms-25-04166-f002:**
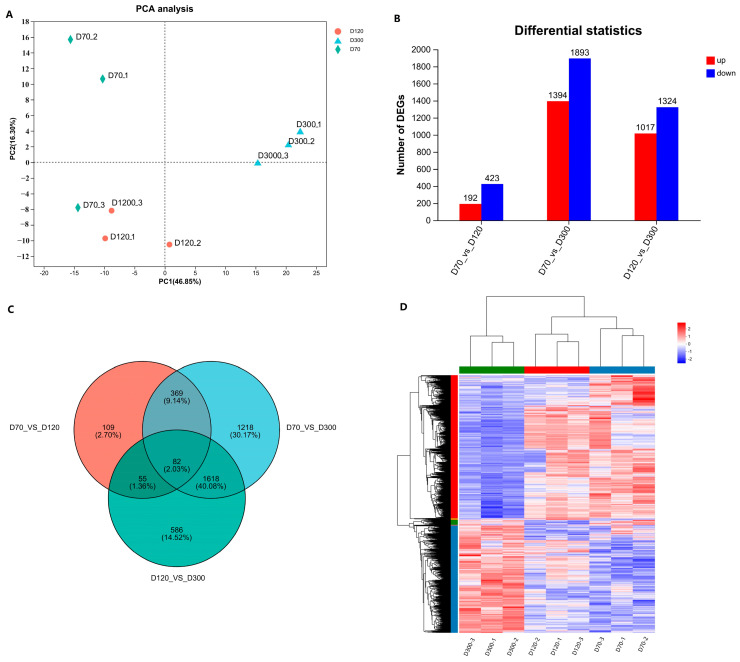
Principal component analysis and screening of DEGs for each sample. (**A**) Analysis of group principal component results. (**B**) The quantitative relationship of DEGs between each group. (**C**) Venn diagram of DEGs. (**D**) Clustering heatmap of DEGs.

**Figure 3 ijms-25-04166-f003:**
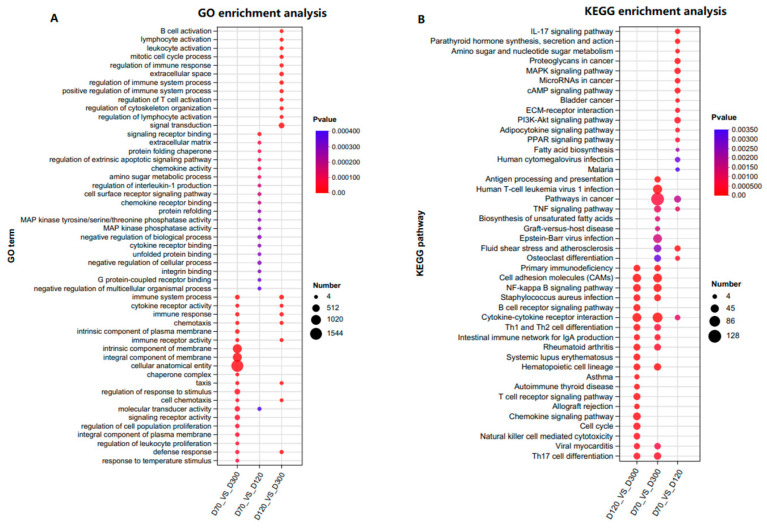
Enrichment functional analysis of DEGs in each comparison group. (**A**) Top 20 GO terms (BP categories) during anserine knob development. (**B**) Top 20 enriched KEGG pathways during goose knob development.

**Figure 4 ijms-25-04166-f004:**
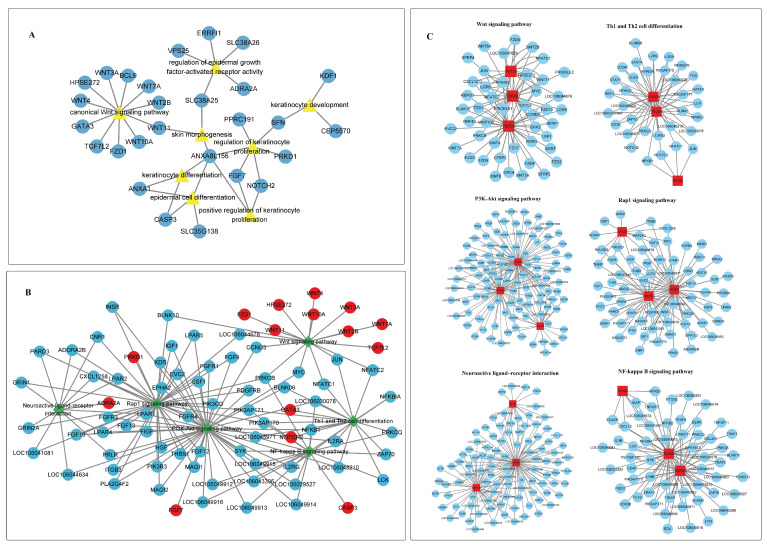
(**A**) Gene network diagram of GO terms during goose knob morphogenesis. (**B**) Gene network map of the KEGG pathway during goose knob morphogenesis. (**C**) DEGs in Wnt, Th1, and Th2 cell differentiation, neuroactive ligand–receptor interaction, NF-kappa B, PI3K-Akt, and Rap1 signaling pathways during the morphogenesis of goose knob.

**Figure 5 ijms-25-04166-f005:**
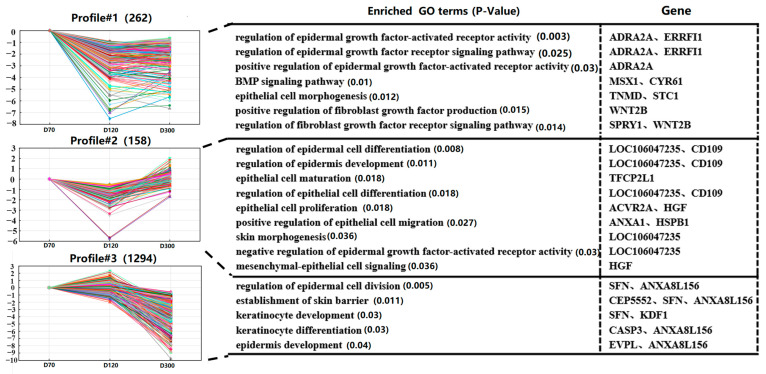
Time series expression analysis of DEGs. Selected clusters of DEGs correspond to biological processes (BPs), and candidate genes are shown next to each cluster. Note: profile#1: clusters1; profile#2: clusters2; profile#3: clusters3.

**Figure 6 ijms-25-04166-f006:**
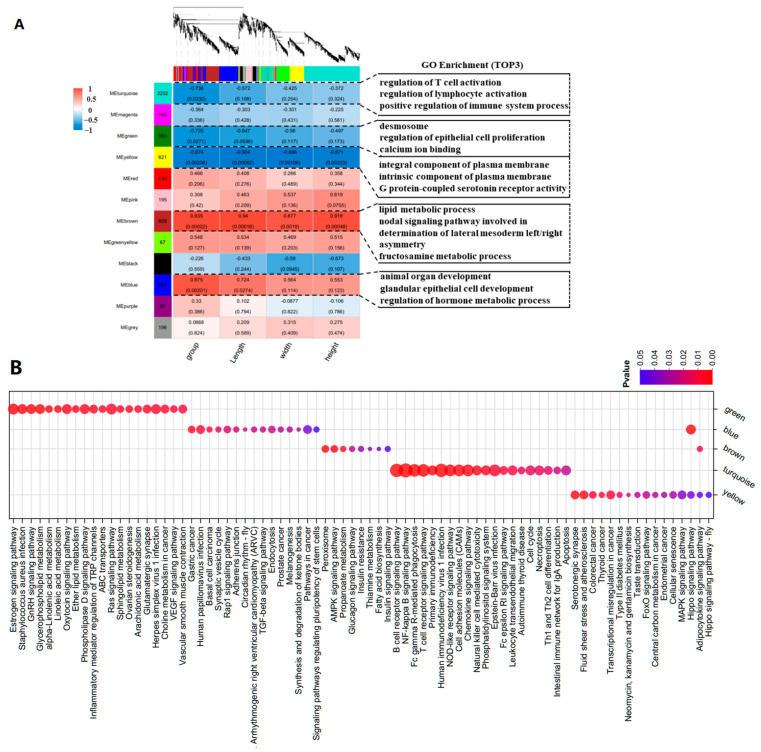
WGCNA analysis. (**A**) Division of gene modules and correlation between gene modules and sample information. The figure shows the clustering of genes, the division of gene modules, and the correlation between gene modules and module information, as well as the GO enrichment (top 3) analysis of modules. (**B**) KEGG functional analysis of key modules. Note: In Figure 6A group (D70, D120, D300), Length represents goose knob height, Width indicates goose knob width, and Height corresponds to goose knob height.

**Figure 7 ijms-25-04166-f007:**
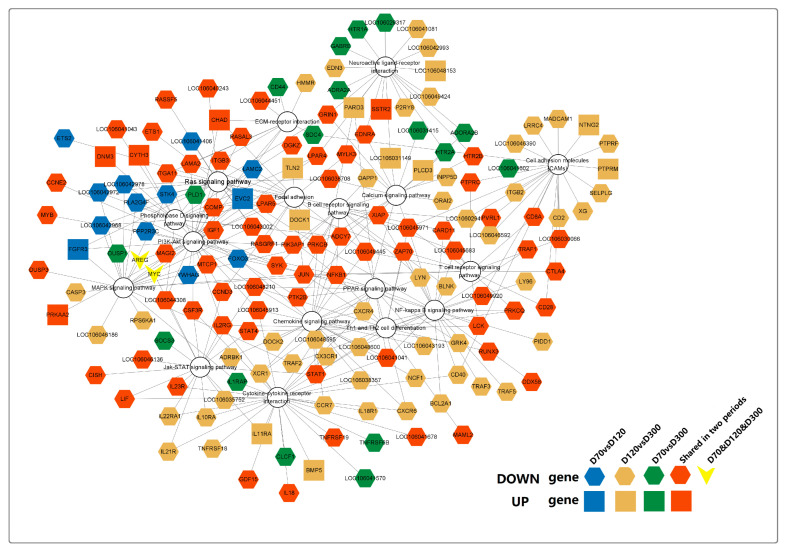
Visualization of pathways involved in goose knob development. This network involves a total of 17 signaling pathways and 167 pathway genes (Pearson correlation ≥ 0.9).

**Figure 8 ijms-25-04166-f008:**
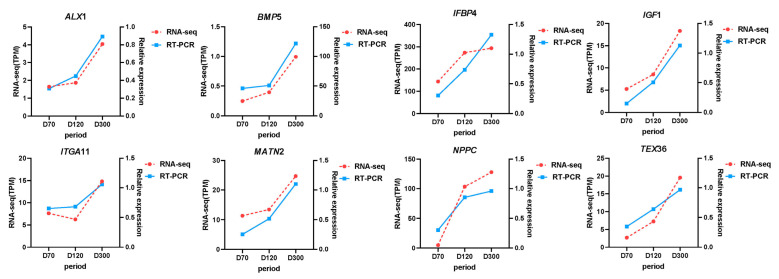
RT-PCR and RNA-seq. The mRNA expression of goose knob development-related genes and four DEGs.

**Figure 9 ijms-25-04166-f009:**
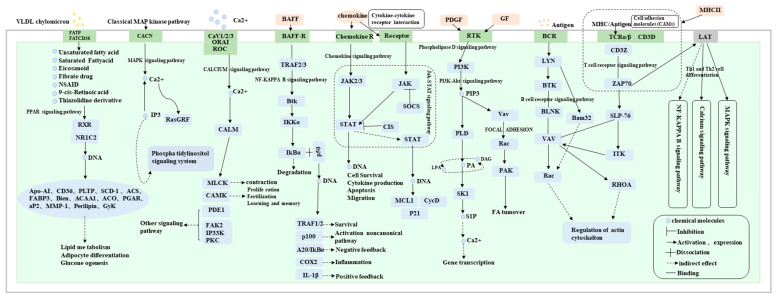
Illustrative summary diagram depicting the regulation of pathways associated with the growth and development of goose knobs.

## Data Availability

The data presented in this study are available upon request from the corresponding author.

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
