# Peer review of "Transcriptome Profiling Unveils Key Genes Regulating the Growth and Development of Yangzhou Goose Knob"

_ijms, 2024, doi:10.3390/ijms25084166_

Round 1

Reviewer 1 Report

Comments and Suggestions for Authors

The presented manuscript is interesting and can be a valuable source of new data regarding genes that determined the growth and development of goose knob. The manuscript used methods that are well-designed and appropriate to conduct this type of research. The authors used the screening RNA-seq approach and proposed the molecular pathways related to the growth of goose knob, which can be a basis for the genetic improvement of this species in the future.

The limitation of the present study is the low number of individuals in each group (n=3). Nevertheless, the qPCR validation confirmed the obtained RNA-seq data thus I think that group size is acceptable.

Specific comments:

Figures 2-6  and 9– please add the software name and version used for Figure preparation

Line 253-257 – different font

Line 398 – show the RIN values obtained for RNA samples

Line 399- did you ligate the libraries in different indexes and ten sequenced together in one pool?

Line 400 - there are lack of methods used for cDNA libraries evaluation

Line 414 - which reference genome was used for GO and KEGG analyses?

Line 433 - the number of technical replicates used in each qPCR analysis should be shown

Table S12 – add the accession number of analyzed DEGs and the amplicon’s length

Results and Discussion section – In my opinion, these sections were clarified and interesting prepared. The conclusion is based on the results obtained.

Author Response

  1. Figures 2-6 and 9– please add the software name and version used for Figure preparation

Response: We gratefully appreciate for your valuable comment. Based on your suggestion, we have incorporated the names and versions of the software used for drawing into the materials and methods section, and reorganized it for clarity. (Lines 434-460, page 14)

Our modifications are as follows :“RSEM software was used to obtain read counts for each sample gene [34]. TPM con-version was performed to obtain standardized gene expression levels and perform principal component analysis (PCA) through the gmodels package. The DESeq2 package was utilized to compute DEGs based on the read count expression of each sample [35]. DEGs with an absolute fold change |log2FC| ≥ 1 and an adjusted P value (P adjust) of < 0.05 were considered to be statistically significant. Excel (2019) was employed to depict the quantitative relationship within each group. The ggVennDiagram package was used to illustrate the intersection genes of DEGs between each group, while the pheatmap package facilitated visualization of the clustering among samples. The GO and KEGG databases were utilized for functional annotation analysis of all sequenced genes, com-pared against the background of the entire transcriptome. Enrichment analysis of DEGs was con-ducted using the Goatools and KOBAS software programs [36-37]. Key pathways were identified through the expression profiles of DEGs within the enriched pathways. Visualization of gene network relationships was achieved using Cytoscape (v3.9.1) soft-ware. Furthermore, time-series expression analysis was performed using the STEM soft-ware to observe gene expression patterns across multiple time points. Subsequently, functional class enrichment analysis was conducted on genes exhibiting specific expression patterns to further identify key genes associated with traits [38-39]. The WGCNA package was used to construct a co-expression network for gene expression and the dynamic tree cutting method (setting the MEDiss Thres cutting line to 0.25 to merge similar modules) was employed to divide the modules to determine the final number of modules. Key mod-ules were screened out through |cor| > 0.7 and P < 0.02 and Correlation analysis was conducted between genes with cor > 0.9 in the key modules and candidate genes related to knob development to determine the key core genes. Cytoscape software was used to visualize all the key pathway networks finally identified related to the development of goose knob. Finally, the regulatory relationships of genes in pathways related to goose knob development are summarized (Power Point-2019).”

  1. Line 253-257 – different font

Response: We feel sorry for the inconvenience brought to the reviewer. We have adjusted the font according to the suggestions of the reviewers. (Lines 266-273, page 9)

  1. Line 398 – show the RIN values obtained for RNA samples

Response: Thank you for pointing out this problem in manuscript. We have shown the RIN values obtained for RNA samples based on your suggestion. (Lines 411-412, page 13)

Our modifications are as follows :“(RIN≥8.4, total RNA amount≥1ug, concentration≥35ng/μL, 1.8<OD260/280<2.0)”

  1. Line 399- did you ligate the libraries in different indexes and ten sequenced together in one pool?

Response: We feel sorry for the inconvenience brought to the reviewer. We utilized a single sample and a single library, rather than a mixed library comprising all samples.

  1. Line 400 - there are lack of methods used for cDNA libraries evaluation

Response: We gratefully appreciate for your valuable suggestion. We have made changes based on your suggestions. (Lines 414-424, page 13)

Our modifications are as follows :“In summary, the procedure begins with the use of magnetic beads containing Oligo (dT) and polyA to perform A-T base pairing. Following mRNA isolation (Kb) from total RNA, fragmentation buffer is added to randomly fragment the mRNA, which is then screened through magnetic beads to isolate small fragments of approximately 300 bp. Next, the SuperScript Double-Stranded cDNA Synthesis Kit and six-base random primers (random hexamers) are employed to reverse-transcribe single-strand cDNA from mRNA, followed by second-strand synthesis to form a stable double-stranded structure. Subsequently, End Repair Mix is added to the synthesized cDNA for end repair, followed by addition of base A and sequencing adapters. PCR amplification is then performed after size selection of cDNA target fragments of 300 base pairs (bp) using a 2% low-range ultra-agarose gel.”

6.Line 414 - which reference genome was used for GO and KEGG analyses?

Response: Thank you for pointing out this problem in manuscript.  The reference genome used is (Anser cygnoides, GCF 000971095.1), The species of the background gene file used in our GO and KEGG analysis is Anser cygnoides, because through query in the ncbi database, the taxid numbers of chiness goose and Anser cygnoides are both 8845.  Software links: KOBAS (bioinfo.org) and Goatools (https://github.com/tanghaibao/Goatools) (Line 430, page 14).

GOATOOLS :Klopfenstein DV, Zhang L, Pedersen BS, Ramírez F, Warwick Vesztrocy A, Naldi A, Mungall CJ, Yunes JM, Botvinnik O, Weigel M, Dampier W, Dessimoz C, Flick P, Tang H. GOATOOLS: A Python library for Gene Ontology analyses. Sci Rep. 2018 Jul 18;8(1):10872. doi: 10.1038/s41598-018-28948-z. PMID: 30022098; PMCID: PMC6052049.

KOBAS :Mao X, Cai T, Olyarchuk JG, Wei L. Automated genome annotation and pathway identification using the KEGG Orthology (KO) as a controlled vocabulary. Bioinformatics. 2005 Oct 1;21(19):3787-93. doi: 10.1093/bioinformatics/bti430. Epub 2005 Apr 7. PMID: 15817693.

7.Line 433 - the number of technical replicates used in each qPCR analysis should be shown

Response: Thank you for pointing out this problem in manuscript. Based on your suggestion we have added displaying the number of technical replicates used in each qPCR analysis. (Lines 468-469, page 14)

Our modifications are as follows :“(Three technical replicates per sample)”

8.Table S12 – add the accession number of analyzed DEGs and the amplicon’s length

Response: We gratefully appreciate for your valuable comment. We have made changes based on your suggestions, including adding the accession numbers of the analyzed DEGs and specifying the amplicon's length.

Reviewer 2 Report

Comments and Suggestions for Authors

In this study, histomorphological and transcriptomic analyses of goose knobs in D70, D120, and D300 Yangzhou goose revealed differential changes in tissue morphology during the growth and development of goose knobs and the key core genes that regulate goose knob traits. This study provides a reference for understanding the molecular mechanisms of goose knob growth and development. However, some questions should be amended to improve the manuscript.

1. I suggest adding a sentence about the importance of the goose knobs in the abstract.

2. L54-70: Since the research subject of this manuscript is goose, the existing research on goose should be described more, and the content of other species can be reduced appropriately. We still don't know whether there is any previous research on goose knobs and what problems still exist at present.

3. L70-71: Are goose knobs species?

4. L73-75: Do various phenotypic traits that have been revealed their genetic basis also include goose knobs?

5. L120-121: How many of the 82 differentially expressed genes are up and down? These genes are important because they may play a role throughout the entire growth and development process of goose knobs.

6. Figure 2: I suggest changing the names of the samples in figures A and D to D70_1, D120_1, D300_1, etc.

7. The resolution of several figures in the manuscript is too low to be seen clearly.

8. L437: Please give reference to the ddCt method.

9. There's a problem with the references section.

Author Response

  1. I suggest adding a sentence about the importance of the goose knobs in the abstract.

Response: We gratefully appreciate for your valuable comment. We have incorporated a sentence regarding the significance of goose knobs, as per your suggestion. (Lines 15-17, page 1)

Our modifications are as follows :“Goose is one of the most economically valuable poultry species and has a distinct appearance due to its possession of a knob. A knob is a hallmark of sexual maturity in goose (Anser cygnodies) and plays crucial roles in artificial selection, health status, social signaling, and body temperature regulation.”

  1. L54-70: Since the research subject of this manuscript is goose, the existing research on goose should be described more, and the content of other species can be reduced appropriately. We still don't know whether there is any previous research on goose knobs and what problems still exist at present.

Response: We gratefully appreciate for your valuable comment. We have implemented your suggestion to include a description of the goose and existing research, along with additional research on the goose knob. (Lines 71-86, page 2)

Our modifications are as follows :Knobs represent a unique appearance characteristic of geese, but their tissue morphology and genetic mechanisms have not yet been revealed. A comprehensive analysis of Lander, Lionhead, and Sichuan white geese using histomorphology, transcriptome, and whole-genome resequencing has, for the first time, unveiled the possibility that DI02 plays a key role in the phenotype of goose knobs. [16]. Through phenotypic observation, it was found that Yangzhou geese with larger knobs at the same age tended to be heavier and more favored by consumers. Therefore, a joint analysis of histomorphology and transcriptomics was conducted to identify BMP5, DCN, TSHR, and ADCY3 as key gene influencing the development of Yangzhou goose knob [17]. These findings indicate that studying head phenotypic features is crucial for understanding growth rate, skull development, and disease observation. Although studies have identified important genes that may affect the phenotype of goose knob, there are currently only the above two relevant reports on the genetic mechanisms affecting goose knob. therefore, the genetic mechanisms affecting the growth and development of goose knob cannot be fully revealed. However, as goose is one of the most economically valuable poultry, it is of great significance to study the genetic basis of goose knob.

  1. L70-71: Are goose knobs species?

Response: We gratefully appreciate for your valuable comment. The knob is formed through long-term selective breeding of Chinese geese (Anser cygnodies) and serves as a sexual marker of maturity in this species. In contrast to European geese, Anser cygnodies exhibit slender necks, and upon reaching adulthood, develop a fleshy bulge on the top of their beaks, commonly referred to as a knob. This distinctive feature sets Anser cygnodies apart from European geese. Typically, at 70 days of age, the forehead is flat with no bulges. By 120 days, it evolves into a mound-shaped bulge with two dimples, eventually maturing into a spherical bulge by 300 days. The size of the knob is larger in male geese compared to females, and male geese with larger knobs also tend to be larger in size.

  1. L73-75: Do various phenotypic traits that have been revealed their genetic basis also include goose knobs?

Response: We gratefully appreciate for your valuable comment. In previous studies,A comprehensive analysis of Lander, Lionhead, and Sichuan white geese using histomorphology, transcriptome, and whole-genome resequencing has, for the first time, unveiled the possibility that DI02 plays a key role in the phenotype of goose knobs. [16]. Through phenotypic observation, it was found that Yangzhou geese with larger knobs at the same age tended to be heavier and more favored by consumers. Therefore, a joint analysis of histomorphology and transcriptomics was conducted to identify BMP5, DCN, TSHR, and ADCY3 as key gene influencing the development of Yangzhou goose knob [17].

Deng, Y.; Hu, S.; Luo, C.; Ouyang, Q.; Li, L.; Ma, J.; Lin, Z.; Chen, J.; Liu, H.; Hu, J.; et al. Integrative analysis of histomorphology, transcriptome and whole genome resequencing identified DIO2 gene as a crucial gene for the protuberant knob located on forehead in geese. BMC Genomics 2021, 22, 487, doi:10.1186/s12864-021-07822-9.

       Ji, W.; Hou, L.E.; Yuan, X.; Gu, T.; Chen, Z.; Zhang, Y.; Zhang, Y.; Chen, G.; Xu, Q.; Zhao, W. Identifying molecular pathways and candidate genes associated with knob traits by transcriptome analysis in the goose (Anser cygnoides). Sci Rep 2021, 11, 11978, doi:10.1038/s41598-021-91269-1.

  1. L120-121: How many of the 82 differentially expressed genes are up and down? These genes are important because they may play a role throughout the entire growth and development process of goose k (Lines 134-135, page 3)

Response: We are sincerely grateful for your valuable suggestion. Based on your suggestions, we have added 82 DEGs (up-regulated: 30; down-regulated: 52).

  1. Figure 2: I suggest changing the names of the samples in figures A and D to D70_1, D120_1, D300_1, etc.

Response: We gratefully appreciate for your valuable suggestion. We have renamed the sample names in Figures A and D to D70_1, D120_1, D300_1, etc., as per your suggestions. (Line 139, page 4)

  1. The resolution of several figures in the manuscript is too low to be seen clearly.

Response: Thank you for pointing out this problem in this manuscript. We have adjusted the resolution based on your suggestions and We have packaged and uploaded the clear version of the original image.

  1. L437: Please give reference to the ddCt method.

Response: We gratefully appreciate for your valuable suggestion. We have cited relevant literature on fluorescence quantification using the 2-ΔΔCt method to calculate the relative expression of genes in the article. (Lines 603-604, page 18)

 Livak, K.J.; Schmittgen,T.D. Analysis of relative gene expression data using real-time quantitative PCR and the 2(-Delta Delta C(T)) Method. Methods. 2001, Dec;25(4):402-408. doi: 10.1006/meth.2001.1262.

  1. There's a problem with the references section.

Response: We feel sorry for the inconvenience brought to the reviewer. We have rechecked the references based on your suggestions and made corrections